# The Effectiveness of Robot- vs. Virtual Reality-Based Gait Rehabilitation: A Propensity Score Matched Cohort

**DOI:** 10.3390/life11060548

**Published:** 2021-06-11

**Authors:** Emilia Biffi, Elena Beretta, Fabio Alexander Storm, Claudio Corbetta, Sandra Strazzer, Alessandra Pedrocchi, Emilia Ambrosini

**Affiliations:** 1Scientific Institute, IRCCS “Eugenio Medea”, Bosisio Parini, 23842 Lecco, Italy; elena.beretta@lanostrafamiglia.it (E.B.); fabio.storm@lanostrafamiglia.it (F.A.S.); claudio.corbetta@lanostrafamiglia.it (C.C.); sandra.strazzer@lanostrafamiglia.it (S.S.); 2NEARLab, Department of Electronics, Information and Bioengineering, Politecnico di Milano, 20133 Milan, Italy; alessandra.pedrocchi@polimi.it

**Keywords:** robot-assisted gait training, virtual reality plus treadmill training, children, acquired brain injury, propensity score matching, cohort study

## Abstract

Robot assisted gait training (RAGT) and virtual reality plus treadmill training (VRTT) are two technologies that can support locomotion rehabilitation in children and adolescents affected by acquired brain injury (ABI). The literature provides evidence of their effectiveness in this population. However, a comparison between these methods is not available. This study aims at comparing the effectiveness of RAGT and VRTT for the gait rehabilitation of children and adolescents suffering from ABI. This is a prospective cohort study with propensity score matching. Between October 2016 and September 2018, all patients undergoing an intensive gait rehabilitation treatment based on RAGT or VRTT were prospectively observed. To minimize selection bias associated with the study design, patients who underwent RAGT or VRTT were retrospectively matched for age, gender, time elapsed from injury, level of impairment, and motor impairment using propensity score in a matching ratio of 1:1. Outcome measures were Gross Motor Function Mesure-88 (GMFM-88), six-min walking test (6MWT), Gillette Functional Assessment Questionnaire (FAQ), and three-dimensional gait analysis (GA). The FAQ and the GMFM-88 had a statistically significant increase in both groups while the 6MWT improved in the RAGT group only. GA highlighted changes at the proximal level in the RAGT group, and at the distal district in the VRTT group. Although preliminary, this work suggests that RAGT and VRTT protocols foster different motor improvements, thus recommending to couple the two therapies in the paediatric population with ABI.

## 1. Introduction

Acquired brain injury (ABI), occurred after a period of normal development, is one of the main causes of death and neurologic disability in children after infancy [1]. Motor impairments are common and often require prolonged assistance [2]. One of the primary rehabilitation goals for children and adolescents suffering from ABI is the improvement of walking ability, in terms of pattern, quality, and independence.

In the last years, standard gait rehabilitation has been flanked by technology-based treatments. Among others, robot-assisted gait training (RAGT) is widely used in the gait rehabilitation of adults with different diseases such as stroke, spinal cord injury, and multiple sclerosis [3,4,5,6], as well as of children and adolescents with neuro-motor impairment [7]. The most common gait rehabilitation robots available for the developmental age are exoskeletons. These devices operate mechanically on the human body by means of cuffs connected to the patient’s lower limbs. There are wearable exoskeletons both for overground walking and for walking on a treadmill. Advantages of exoskeletons with respect to standard gait training are the repetitiveness of the movement, the intensity of the treatment, and the possibility to be used in the absence of independent walking. Recent experimental works [8,9] and a meta-analysis [10] concluded that RAGT benefits patients with cerebral palsy (CP) by increasing walking speed and endurance and improving gross motor function. On the other hand, the impact of RAGT on the gait pattern of children with CP is not confirmed [11,12]. Moreover, its advantage compared to conventional therapy is still unclear in this population [13].

Few studies describing the use of RAGT in young patients with ABI are available [14,15,16,17]. These studies suggested that RAGT in this population is effective in terms of endurance, gross motor abilities, as well as gait speed and hip kinematics.

Beside robotic rehabilitation, gait training can take advantage of virtual reality (VR) [18,19]. VR is defined as an artificial environment, where patients are able to experience live interactions with a gaming environment through sensory stimuli and/or motor feedback. VR therefore includes videogaming consoles (e.g., Wii, PlayStation, Xbox Kinect), caves (e.g., Motek Caren), as well as head mounted displays (e.g., Oculus, HTC VIVE). The different degree of immersion determines the state of presence, i.e., the experience that the environment is real [20]. Advantages of VR interventions are the capability to offer a multitude of activities, and to support motivation and engagement, especially in the youngest. This may improve patients’ adherence to training plans because of the youth population’s increasing exposure to electronic games.

A recent meta-analysis has shown the effectiveness of VR for gait rehabilitation in patients with CP, specifically in terms of speed, stride length and gross motor function [18]. Conversely, Fandim and co-authors in their systematic review suggest that there is no benefit of adding VR to conventional rehabilitation of children and young adults with CP, even if the quality of the evidence is low [19].

There is a paucity of literature exploring the use of VR, specifically coupled to treadmill training (VRTT), for the gait rehabilitation of children and adolescents affected by ABI [21,22]. These works, albeit preliminary and limited by small sample size, suggest benefits in terms of gross motor function, endurance, step length, gait speed, and autonomy in daily life activities.

Although rehabilitative protocols including RAGT or VRTT have been proposed in children and adolescents with ABI, no studies have compared the effectiveness of these two rehabilitation technologies in a pediatric population affected by ABI and there are no guidelines supporting the use of one or the other.

Prospective, double-blind, randomized clinical trials are the gold standard to evaluate the efficacy of a healthcare intervention in a well-designed target population. However, observational data sets also contain useful information to evaluate daily clinical practice, although affected by selection bias and confounding factors [23,24,25]. Propensity-score matching (PSM) was first described in 1983 [26] as a technique to reduce bias from confounding variables at baseline. PSM attempts to mimic randomization on observed covariates: the propensity score is estimated using logistic regression and represents the likelihood that a subject will be included in an intervention group or the other based on an observed set of baseline covariates (such as demographic, socioeconomic state and clinical characteristics). Thus, subjects with the same propensity scores in the two intervention groups have identical distributions for all the observed covariates [26].

The aim of this work was to compare the effectiveness of two technology-based interventions for gait rehabilitation in children and adolescents suffering from ABI. One intervention was based on RAGT and the other on VRTT, and both were coupled with standard physiotherapy. The study design was a cohort study. Being observational, a propensity score matching algorithm was retrospectively used to match the two groups and compare the two interventions. The final aim of this work was to suggest possible guidelines to select the best treatment for these patients, thus improving their motor outcomes and, eventually, daily life.

## 2. Methods

### 2.1. Participants

Between October 2016 and September 2018 all consecutive patients undergoing an intensive gait rehabilitation treatment at the Scientific Institute E. Medea (Bosisio Parini, Italy) assisted by a robot or using a treadmill plus virtual reality system were prospectively observed. Inclusion criteria were: diagnosis of ABI; age between 4 and 20 years; a level of motor impairment ranging from I to IV, classified according to the Gross Motor Function Classification System (GMFCS); adequate comprehension and cooperation; and absence of visual impairment. Exclusion criteria were: severe muscle spasticity; injection of botulinum toxin in lower limbs during the 6 months prior to the enrollment; variation in oral skeletal muscle relaxant drug dose in the month prior to treatment; previous orthopedic surgery; a diagnosis of severe learning disabilities; behavioral problems; visual or hearing difficulties that would impact on function and participation.

This study was performed in accordance with the Declaration of Helsinki and the Ethics Committee of Scientific Institute E. Medea approved the observational study protocol (protocol code: GIP355; date of approval; 23 September 2016). Patients or their parents provided written informed consent. The trial was registered in the repository of the Italian Ministry of Health (registration number: 001095).

### 2.2. Intervention

The intervention lasted one month and consisted of 20 45-min sessions of conventional physiotherapy and 20 45-min sessions of either RAGT or VRTT. Being an observational study, patients were assigned to RAGT or VRTT intervention on the basis of clinical decisions.

The conventional physiotherapy included stretching of hip flexor and hamstring muscles, muscle strengthening exercises, such as squats, static and dynamic balance training, postural transitions such as sit-to-stand, and over ground walking training with particular attention to gait smoothness, stability, and endurance.

#### 2.2.1. Robot-Assisted Gait Training

RAGT was performed using the Lokomat^®^ (Hocoma AG, Volketswil, Switzerland), an active lower limb exoskeleton (Figure 1A). During training, speed, body-weight support and guidance force were personalized on each patient to assure active participation. The initial body-weight support was set at 50%, and gradually decreased according to the individual’s response to the intervention. The guidance force was initially set to 100%, and then gradually reduced up to 5% above the automatic stop threshold. To engage subjects and to increase their active participation and motivation in gait practice, therapists provided frequent oral encouragement and augmented performance feedback (implemented in the exergames) was used in all the sessions A therapist, trained and certified by Hocoma, was always present during the training sessions.

#### 2.2.2. Virtual Reality Plus Treadmill Training Gait Rehabilitation

The VRTT was performed with the Gait Real-time Analysis Interactive Lab (GRAIL, Motek, Houten, The Netherlands) that is an immersive VR system for gait assessment and rehabilitation (Figure 1B). It is equipped with a dual-belt treadmill, a two-degree of freedom platform, and a 180° cylindrical screen where virtual environments are projected and synchronized with the treadmill and the subject. A Vicon motion-capture system (Oxford Metrics, Oxford, UK) equipped with 10 optoelectronic cameras (sample frequency 100 Hz) surrounds the system. Subjects interact with virtual environments with their movement, thanks to passive markers located in different body parts depending on the activity. The system returns visual, proprioceptive and auditory feedback to the subject to support rehabilitation.

The VRTT included exercises to improve walking and balance abilities in engaging VR environments, for example, by displaying in real-time the joints kinematic during walking through a forest or by transferring load from one body side to the other to avoid obstacles while practicing ski. The training was highly personalized for the motor and cognitive performance of each patient. Experienced physiotherapists, trained and certified by Motek, defined and performed the training sessions on the GRAIL system.

### 2.3. Assessment

Baseline measures included: patient’s age at the beginning of the therapy and at occurrence of the ABI, time elapsed from injury, gender, etiology, motor impairment, intelligence quotient (IQ) and Gross Motor Function Classification System (GMFCS) level. GMFCS is a 5-level classification system describing the gross motor function of children and adolescents [27], and has been previously used to classify subjects with ABI [28].

Participants underwent a motor assessment before (T0) and at the end of the treatment (T1), which included the following outcome measures: Gross Motor Function Measure-88 (GMFM-88), which was selected as primary outcome, 6 min walking test distance (6MWT), Gillette Functional Assessment Questionnaire (FAQ), and 3-dimensional gait analysis (GA).

The GMFM-88 is an assessment tool designed for the assessment of gross motor function in children and adolescents (under 18 years old) with cerebral palsy and includes 88 items, divided into 5 dimensions, each of them representing a particular movement or position. Items span the spectrum of gross motor activities in five dimensions: A: Lying and rolling; B: Sitting; C: Crawling and kneeling; D: Standing; E: Walking, running, and jumping. Total score and dimensions D and E were considered in this study, since more related to the interventions. The validity of GMFM-88 in the evaluation of gross motor function in children with ABI has been previously demonstrated [29].

The 6MWT rates gait endurance during self-paced walking within 6 min through the hospital corridors. Verbal standardized instructions are given to the patient during the test, which includes walking at a comfortable speed, turning 180° every 25 m and covering as much distance as possible within the time limit of 6 min [30].

The FAQ is a questionnaire that assesses levels of mobility during everyday life, in a 10-level classification [31]. The FAQ is administered by asking questions to the parents or the child him/herself.

GA performs a quantitative analysis of gait movement. The GA laboratory is equipped with eight optoelectronic cameras, an optoelectronic system (Elite, BTS Bioengineering, Milan, Italy) with a sampling rate of 100 Hz, and two force plates (Kistler Group, Winterthur, Switzerland) embedded in the floor. Patients were asked to walk at their preferred speed, and to wear their orthoses and footwear only if they were unable to walk barefoot.

### 2.4. Data Analysis and Statistics

#### 2.4.1. The Propensity Score Algorithm

A PSM algorithm was used to identify matched cohorts as a subgroup of the unmatched cohorts, and was defined as follows. First, covariates were selected among the baseline measurements under the hypothesis that they contribute to the choice of the treatment. Age, gender, time elapsed from injury, GMFCS and motor impairment were selected as covariates. Then, a logistic regression was performed to estimate the propensity scores, considering the intervention as outcome variable and selected covariates as predictors. The matching between RAGT group and VRTT group was obtained by using the 1:1 nearest-neighbor procedure that means that each individual of the VRTT group was matched with one of the RAGT group in terms of propensity score, discarding individuals with propensity scores outside the range of the other group. Finally, to check the model adequacy, the standardized differences between the groups were computed before and after matching for continuous, dichotomous, and categorical variables, according to [25]. The PSM algorithm was developed in Rstudio by means of the MatchIt library. The matchit() function was used with the method “nearest” to implement the 1:1 nearest-neighbor matching.

#### 2.4.2. Gait Parameters Extraction

For each GA assessment, an expert physiotherapist collected and processed at least five trials for the left and the right limbs using dedicated software (EliteClinic, BTS Bioengineering, Milan, Italy). The most representative trial was then selected for further analyses. BTS Smart Clinic software was used to extract spatio-temporal and kinematic data for each selected gait cycle.

Spatiotemporal features included: walking velocity, cadence, bilateral stride duration, and bilateral step length and width.

Kinematic curves were analyzed in Matlab by using an ad hoc algorithm designed to extract, for the right and left leg, the foot progression in stance, maximum and minimum flexion angle and the range of motion (ROM) in the sagittal plane for ankle, knee, and hip, and the ROM of pelvic tilt, obliquity, and rotation.

Furthermore, starting from kinematic data, the BTS Smart Clinic software automatically computed the Gait Deviation Index (GDI). The GDI was developed and validated by Schwartz and Rozumalski in 2008 [32]. It is defined as the scaled distance between 15 gait feature scores (selected as those that explain the 98% of data) for a subject and the average of the same 15 gait feature scores for a control group of typically developing children. Therefore, the GDI provides an overall assessment of the deviation from a physiological gait pattern. The GDI ranges from 0 to 100, where 100 indicates the absence of gait pathology [32].

For the GA parameters, the mean value between left and right side was considered.

#### 2.4.3. Statistics

The Kolmogorov–Smirnov test was run to test data distribution; since normality was not verified, non-parametric tests were used, and data were represented with median and interquartile range values.

A Mann–Whitney U test and a Pearson Chi-squared test were performed between groups on continuous and dichotomous/categorical baseline measures, respectively, before and after the propensity score matching.

Considering the matched cohorts, the time effect was evaluated independently in each group by comparing baseline and post-treatment scores by means of the Wilcoxon signed rank test. The effect of the intervention (group effect) was evaluated by comparing the pre-post changes of each outcome between the two groups using the Mann–Whitney U test.

Finally, when the minimal clinical important difference (MCID) was available in the literature, the percentages of patients who exhibited a clinically important change (pre-post improvement above the MCID) were computed for the two groups. Similarly, the percentages of patients experiencing a worsening above the MCID were computed. A Pearson Chi-squared test was performed between groups to look for differences in terms of improved, stable and worsened patients. For 6MWT and GMFM-88 (and its items D and E), MCID values were set at 30 m and 5, respectively, as suggested by [33]. For step length, MCID was defined equal to 0.2 m as defined by [34], while for gait kinematics in the sagittal plane MCID was set at 5° as proposed in [35].

The statistical analysis was performed in SPSS v21. The significance level was established at *p* < 0.05.

## 3. Results

### 3.1. Unmatched Cohort

The unmatched cohort was composed of 70 patients allocated into two groups in a non-randomized way: 39 were allocated in the RAGT group and 31 were allocated in the VRTT group. The IQ evaluation was available for 57 patients (28 in the RAGT group, 29 in the VRTT group). Due to equipment availability, or the inability of patients to perform a test, 6MWT was performed in 32 patients in the RAGT group and 28 patients in the VRTT group, FAQ was present for 38 patients in the RAGT group and 15 patients in the VRTT group, GMFM was performed in 34 patients in the RAGT group and 27 patients in the VRTT group, and GA was available for 20 patients in the RAGT group and 26 patients in the VRTT group. There were no dropouts in the study (Figure 2).

Baseline measures in the unmatched cohorts are shown in Table 1. Differences between groups were found in the time elapsed from injury, severity of the impairment, motor impairment and etiology. Gender, age, and IQ did not show significant differences between the two groups.

### 3.2. Matched Cohorts

The PSM algorithm identified 15 patients in each group. The median and interquartile range of the baseline measures in the matched cohorts are shown in Table 2. No statistically significant differences in any of the baseline variables were observed.

The standardized differences of baseline measurements before and after the matching is shown in Figure 3. The SD after the match was reduced (mean value 0.7 ± 0.6 before, 0.4 ± 0.3 after the matching procedure), except for age at injury and gender, which were already quite small in the unmatched cohort.

### 3.3. Outcomes in Matched Cohorts

Figure 4 shows GMFM, 6MWT, and FAQ in the matched groups, before and after treatment. Both groups presented statistically significant improvement for the primary outcome, with the GMFM-88 increasing in RAGT group (Wilcoxon signed rank test *p* = 0.003) as well as in VRTT group (*p* = 0.009). Furthermore, both groups showed statistically significant improvements in GMFM dimensions D (Wilcoxon signed rank test *p* = 0.005 for RAGT and *p* = 0.018 for VRTT) and E (Wilcoxon signed rank test *p* = 0.002 for both RAGT and VRTT). The percentage of patients with clinically relevant changes in GMFM-88, GMFM-D and GMFM-E were 54%, 62%, and 69% in the RAGT and 21%, 29% and 50% in the VRTT group. Nobody experienced a worsening in his/her gross motor abilities. The FAQ significantly increased in both groups (Wilcoxon signed rank test *p* = 0.017 for RAGT and *p* = 0.046 for VRTT), while the 6MWT improved significantly in the RAGT group (Wilcoxon signed rank test *p* = 0.003) with 53% of patients with clinically relevant changes and 7% of patients with a worsening above MCID and had a trend of improvement (*p* = 0.056) in the VRTT group, with 43% of patients with clinically relevant improvements and 0% with clinically relevant worsening. No differences in the therapy effect were found, as demonstrated by the Mann–Whitney U test (all *p*-values > 0.070).

Table 3 shows spatiotemporal parameters in the matched cohorts: step length and stride length significantly improved only in the RAGT group. However, the percentage of patients with clinically relevant changes in the step length was equal to 8% in both groups. In contrast, the GDI had a statistically significant improvement only in the VRTT group. No statistically significant differences between the two groups were found in any of the analyzed parameters, as shown by the last column that reports the pre-post change for each group and *p*-values obtained with Mann–Whitney U test.

The kinematic measures evaluated with the GA, highlighted that each treatment targeted different joints (see Table 4). Specifically, the VRTT group experienced improvement at the foot and ankle level. The minimum ankle flexion improved above MCID in 31% of patients in the VRTT group and in 25% in the RAGT group. ROM of ankle flexion showed clinically relevant changes in 23% of patients in the VRTT group while in the RAGT group 17% of patients had clinically relevant improvement and 8% had clinically relevant worsening. Considering the minimum knee flexion, the VRTT group showed a significant worsening, with 0% of patients improving and 15% getting worse. However, the same parameter improved in 8% of patients and worsened in 33% of patients in the RAGT group. In contrast, ROM of knee flexion significantly improved in the RAGT group, with 42% of patients improving above MCID, while 23% of patients improved and 8% worsened in the VRTT group. RAGT group significantly improved also in the ROM of hip flexion, with 33% of patients above MCID (vs. 23% in the VRTT group) and 8% with a significant deterioration, the same worsening obtained in the VRTT group. Considering the pelvis, only the RAGT group showed statistically significant improvements. No significant differences between the two interventions were found, as shown by the group effect analysis, reported in the last column.

Table 5 reports the number of improved, stable, and worsened patients in each group. No significant differences between the two groups were found.

## 4. Discussion

The main aim of this study was to compare two different interventions for gait rehabilitation in children and adolescents with ABI, one exploiting RAGT using the Lokomat device and one using immersive VRTT with the Grail system. The assignment of a patient to a treatment depended on clinicians’ decisions and thus provided unmatched cohorts. Therefore, a retrospective matching procedure was mandatory before comparing the efficacy of the two interventions.

Data analysis performed on the matched cohorts revealed that gross motor abilities significantly improved in both groups. However, specifically considering the GMFM-88 and GMFM-D there is a trend of higher percentage of patients in the RAGT group that gained a clinically relevant change. Results also showed that similar percentages of patients in each group improved their endurance and their step length above MCID, even though the improvements were statistically significant only for the RAGT group. Therefore, patients who underwent RAGT treatment had a slightly higher functional gain. Regarding gait analysis, data confirmed a beneficial intervention of RAGT at proximal level (i.e., pelvis and hip) and a positive effect of VR on distal districts (i.e., foot and ankle) and on the overall gait pattern quality. Both treatments barely worked on knee joint. The statistical analysis did not find any differences between the two interventions.

These results are in accordance with previous studies describing the effectiveness of RAGT and VRTT on the locomotion of children with ABI. Indeed, previous studies on RAGT described a proximal-to-distal differential effect on the lower limbs [15] and an enrichment of the main functional measures [14,16]. Interestingly, in the current work, no changes were observed in terms of gait speed and knee district, with only a small percentage of patients showing improvements in terms of knee range of motion. This may be due to differences in the participants’ severity: the investigated cohorts included in this work were composed by patients with mild impairment (i.e., 28 patients with GMFCS II, one patient with GMFCS I and only one patient with GMFCS III) while previous works by Beretta and collaborators included also GMFCS III and IV, i.e., with more severe motor impairment and thus with more potential for improvement. Furthermore, this work confirms results obtained in previous preliminary studies showing the effectiveness of VR on gross motor abilities and on distal joints of children and adolescents suffering from ABI [21,22].

The results of this work suggest that RAGT treatment and VRTT treatment are both effective although working on different districts and competencies. This could find an explanation in the type of intervention performed with the two devices. On one hand, exoskeletons for the lower limbs provide several repeated movements with a fixed kinematic trajectory: this is of course is beneficial in terms of endurance but eliminates variations in the kinematics, which are fundamental for therapy-mediated motor re-learning. Therefore, the reduced sensory feedback might explain the small improvement of the gait pattern [36]. Furthermore, no improvements at distal level were observed in the RAGT group and this can be explained by the fixation of the ankle joint. On the other hand, VRTT is not susceptible to exoskeleton constraints and provides training of the lateral weight shift considering the natural variability in leg and pelvis kinematics. Furthermore, it allows for a task-oriented training that focuses on the practice of skilled motor performance (i.e., locomotion), fostering neural reorganization [37]. These may lead to an increased room for improvement in locomotion pattern.

This work has some limitations. First, although the initial cohorts were quite large compared to traditional studies involving RAGT or VR rehabilitation in the developmental age, the use of the PSM caused a reduction of the sample size, which was small in the matched cohorts (N = 30). Indeed, the PSM has been used in several studies with large sample sizes by matching cases between groups (see as examples [38,39]), but only once in a small sample size [40]. Nevertheless, PSM is a powerful tool that enables excellent matching of baseline characteristics and thus mimics randomization.

A second limitation is that standardized differences remained moderate after the matching, likely due to the small sample size. However, in small matched samples, moderate SD could still be consistent with a correctly specified PSM [25].

A third limitation is related to the use of PSM. Although adjustment was made for several variables, it is possible that residual confounders between the groups could have been omitted in the analysis. Nevertheless, in this study, many covariates were used in the propensity model thus maximally reducing baseline differences between cohorts.

Another issue is the absence of a follow-up assessment, and therefore we could not observe effects in the medium- or long-term. However, the main goal of this study was to assess possible differences between two gait interventions based on advanced technologies, and a pre/post study is a first step in this direction.

Finally, the range of age of the participants was quite broad and the small sample size did not allow to perform age stratification.

Despite such limitations, the authors believe that this is a valuable and novel work, even if addressing a niche topic, that can provide suggestions and open new perspectives on the use of rehabilitation technologies in the developmental age.

Future works will use larger sample size, will compare the intervention effect at different ages and will investigate long-term benefits. Finally, considering that a percentage (about 10%) of patients in both groups experienced worsening in some variables, it needs to be investigated which patient feature and what environmental component or emotive/psychological aspect determines the response to treatment.

## 5. Conclusions

This work compared the effectiveness of two interventions (i.e., RAGT and VRTT) for the gait rehabilitation of children and adolescents suffering from ABI. To our knowledge, this is the first time that these rehabilitation technologies have been compared in paediatric populations. Recently, more and more rehabilitation technologies have come on the market. Each of them has its indication for use but this is often too generic, e.g., for gait rehabilitation, and does not provide specific indications on target users. Thus, it is difficult for a clinician to choose the best device for each patient. This work would like to contribute in this direction. The approach used and the results obtained, although preliminary, pave the way for the definition of guidelines for the treatment of children and adolescents suffering from ABI. Our observations suggested that RAGT and VRTT protocols foster different motor improvements, with RAGT inducing an improvement in terms of endurance and proximal joint kinematics and VRTT enhancing gait pattern and distal joint kinematics. Therefore, a good approach could be to couple the two interventions in order to achieve a more complete recovery of walking ability.

## Figures and Tables

**Figure 1 life-11-00548-f001:**
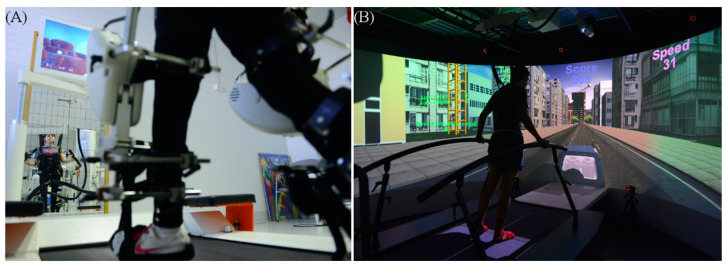
(**A**) The Lokomat device. (**B**) The Grail system.

**Figure 2 life-11-00548-f002:**
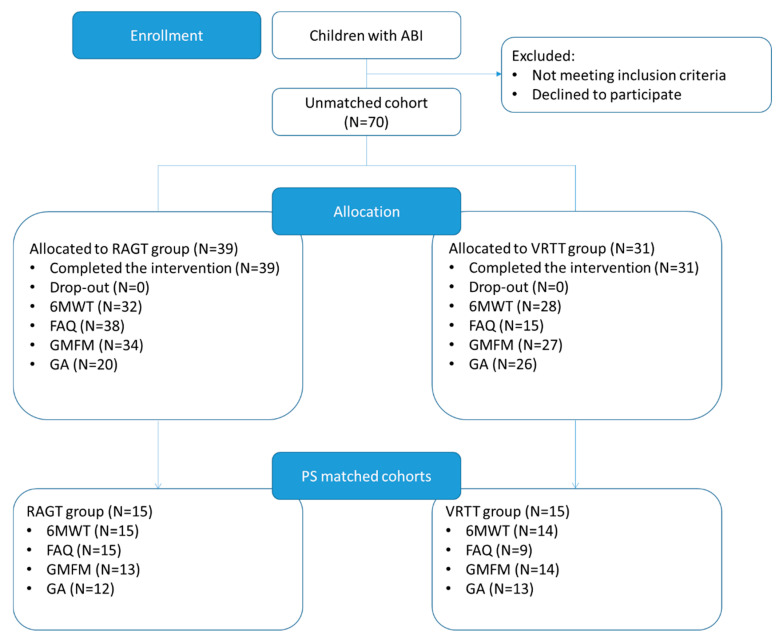
Flowchart of the study. GA: Gait Analysis, N: number of patients.

**Figure 3 life-11-00548-f003:**
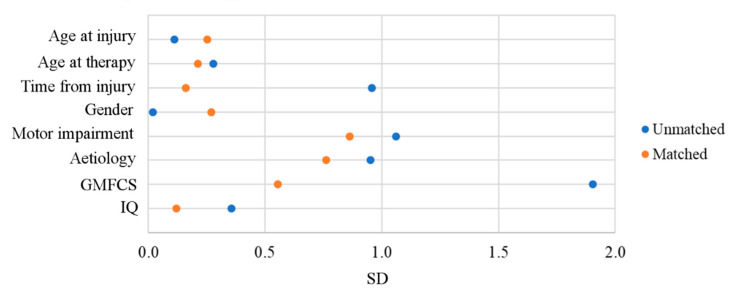
Standardized Mean Differences in the unmatched cohorts (N = 70, blue dots) and after the matching (N = 30, red dots).

**Figure 4 life-11-00548-f004:**
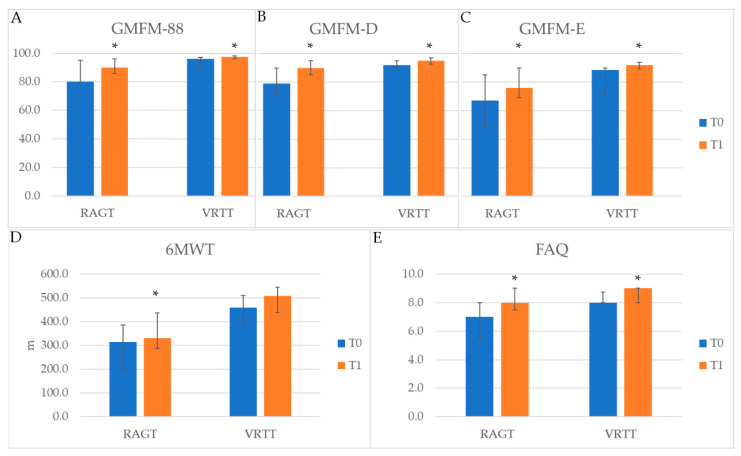
Functional measures of the matched cohorts. * Wilcoxon signed rank test with *p* < 0.05. (**A**) GMFM: Gross Motor Function Measure; (**B**) GMFM-D and (**C**) GMFM-E; (**D**) 6MWT: 6 min walking test; (**E**) FAQ: Gillette Functional Assessment Questionnaire; RAGT: Robot-Assisted Gait Training; VRTT: Virtual Reality Treadmill Training.

**Table 1 life-11-00548-t001:** Median and Interquartile Range of the baseline measures of unmatched cohorts. Statistically significant differences between the two groups are in bold. R: right; L: left; AVM: Arteriovenous malformation; GMFCS: Gross Motor Function Classification System; IQ: Intelligent Quotient; RAGT: Robot-Assisted Gait Training; VRTT: Virtual Reality Treadmill Training. * Median (interquartile range); § Mann-Whitney U test; ‡ Pearson Chi-squared.

	RAGT (N = 39)	VRTT (N = 31)	*p*-Value
Age at injury, years *	12.9 (8.6)	10.6 (7.9)	0.535 §
Age at therapy, years *	13.5 (8.3)	14.6 (5.8)	0.295 §
Time from injury, years *	0.5 (0.5)	2.0 (3.0)	**<0.001** §
Gender (male/female) *	18/21	14/17	0.934 ‡
Motor impairment (tetraparesis/R hemiparesis/L hemiparesis/ataxia/paraparesis/diplegia)	13/11/7/8/0	4/2/6/18/1	**0.004** ‡
Aetiology (head trauma/stroke/tumour /encephalitis/AVM) *	16/7/6/8/2	6/4/16/2/3	**0.011** ‡
GMFCS *	1/13/14/11	4/27/0/0	**<0.001** ‡
IQ *	71.0 (21.8)	77.0 (27.5)	0.093 §

**Table 2 life-11-00548-t002:** Median and Interquartile Range of the baseline measures of matched cohorts. R: right; L: left; AVM: Arteriovenous malformation; GMFCS: Gross Motor Function Classification System; IQ: Intelligent Quotient; RAGT: Robot-Assisted Gait Training; VRTT: Virtual Reality Treadmill Training. * Median (interquartile range); § Mann-Whitney U test; ‡ Pearson Chi-squared.

	RAGT (N = 15)	VRTT (N = 15)	*p*-Value
Age at injury, years *	14.7 (9.7)	9.5 (7.0)	0.507 §
Age at therapy, years *	15.0 (10.6)	11.3 (5.2)	0.604 §
Time from injury, years *	0.5 (1.1)	1.3 (1.3)	0.351 §
Gender (male/female) *	6/9	8/7	0.464 ‡
Motor impairment (tetraparesis/R hemiparesis/L hemiparesis/ataxia/paraparesis/diplegia)	1/7/4/3/0	2/2/4/7/0	0.194 ‡
Aetiology (head trauma/stroke/tumour /encephalitis/AVM) *	3/6/3/2/1	2/3/8/1/1	0.433 ‡
GMFCS *	1/13/1/0	0/15/0/0	0.343 ‡
IQ *	81.0 (18.0)	87.0 (24.0)	0.463 §

**Table 3 life-11-00548-t003:** Gait spatio-temporal parameters of the matched cohorts. Statistically significant comparisons are in bold. h: height; GDI: gait deviation index; IQR: interquartile range; RAGT: Robot-Assisted Gait Training; VRTT: Virtual Reality Treadmill Training. ** Mann-Whitney U test; ~ Wilcoxon signed rank test.

		T0	T1	T0 vs. T1	T1–T0 Change	
	Group	Median (IQR)	Median (IQR)	*p*-Value ~ (Time Effect)	Median (IQR)	*p*-Value ** (Group Effect)
Velocity (ms^−1^)	RAGT	0.55 (0.6)	0.65 (0.35)	0.112	0.1 (0.1)	0.761
	VRTT	0.7 (0.3)	0.8 (0.2)	0.165	0.1 (0.4)	
Velocity/h (s^−1^)	RAGT	34.76 (35.15)	41 (22.02)	0.117	2.48 (6.16)	0.870
	VRTT	55.08 (21.56)	59.48 (16.29)	0.279	5.68 (20.17)	
Cadence (step·min^−1^)	RAGT	92.1 (34.2)	93 (21.9)	0.423	0.6 (8.7)	0.913
	VRTT	108.6 (27)	105.6 (22.2)	0.624	4.2 (19.2)	
Stride time (s)	RAGT	1.31 (0.53)	1.29 (0.30)	0.505	−0.01 (0.13)	0.765
	VRTT	1.12 (0.305)	1.14 (0.24)	0.363	−0.04 (0.23)	
Stance % (0–100)	RAGT	66.01 (5.48)	64.32 (3.08)	0.099	−1.26 (5.2)	0.496
	VRTT	62.84 (4.5)	62.48 (3.83)	0.249	−0.9 (5.38)	
GDI	RAGT	81.9 (8.58)	84.25 (9.11)	0.158	0.83 (2.02)	0.355
	VRTT	85.82 (11.68)	87.87 (11.94)	**0.028**	3.33 (7.79)	
Step width (m)	RAGT	0.16 (0.05)	0.14 (0.06)	0.406	−0.01 (0.04)	0.913
	VRTT	0.15 (0.04)	0.15 (0.03)	0.549	0.01 (0.03)	
Step length (m)	RAGT	0.38 (0.20)	0.41 (0.19)	**0.020**	0.028 (0.05)	0.644
	VRTT	0.41 (0.15)	0.47 (0.09)	0.108	0.06 (0.12)	
Stride length (m)	RAGT	0.76 (0.40)	0.82 (0.39)	**0.023**	0.05 (0.1)	0.724
	VRTT	0.82 (0.31)	0.93 (0.17)	0.124	0.11 (0.24)	

**Table 4 life-11-00548-t004:** Gait kinematic parameters of the matched cohorts. All parameters are in degrees. Statistically significant comparisons are in bold. IQR: interquartile range; RAGT: Robot-Assisted Gait Training; VRTT: Virtual Reality Treadmill Training; ROM: Range of Motion. ** Mann-Whitney U test; ~ Wilcoxon signed rank test; ^a^ maximum dorsiflexion; ^b^ maximum plantarflexion.

		T0	T1	T0 vs. T1	T1–T0 Change	
	Group	Median (IQR)	Median (IQR)	*p*-Value ~ (Time Effect)	Median (IQR)	*p*-Value ** (Group Effect)
Foot progression- Stance	RAGT	−19.5 (7.23)	−19.5 (11.35)	0.695	0.06 (4.27)	0.103
	VRTT	−20.05 (8.05)	−18.0 (11.34)	**0.033**	3.45 (5.09)	
Maximum ankle flexion ^a^	RAGT	11.6 (6.48)	12.63 (7.96)	0.875	−0.33 (3.85)	0.550
	VRTT	7.45 (4.55)	8.8 (6.65)	0.701	−0.40 (5.75)	
Minimum ankle flexion ^b^	RAGT	−12.9 (11.39)	−13.48 (7.79)	0.388	−2.4 (4.02)	0.103
	VRTT	−17.85 (7.05)	−20.95 (10.4)	**0.009**	−4.0 (3.0)	
ROM ankle flexion	RAGT	16.08 (10.9)	17.48 (9.55)	0.432	0.6 (1.05)	0.174
	VRTT	14.8 (5.85)	19.35 (5.65)	**0.023**	3.4 (4.45)	
Maximum knee flexion	RAGT	48.15 (13.0)	51.05 (12.08)	0.388	−0.01 (0.13)	0.870
	VRTT	52.6 (2.75)	55.75 (6.1)	0.753	2.5 (11.69)	
Minimum knee flexion	RAGT	−5.95 (7.01)	−6.8 (4.15)	0.433	−0.48 (7.91)	0.415
	VRTT	−4.7 (6.65)	−8.75 (10.05)	**0.023**	−2 (2.4)	
ROM knee flexion	RAGT	49.9 (15.8)	55.93 (8.6)	**0.010**	3.85 (4.43)	0.480
	VRTT	56.3 (8.75)	58.95 (7.8)	0.075	2.4 (3.1)	
Maximum hip flexion	RAGT	37.68 (4.29)	38.3 (9.29)	0.657	0.75 (4.7)	0.355
	VRTT	31.9 (9.85)	32.05 (9.1)	0.087	−3.1 (3.3)	
Minimum hip flexion	RAGT	0.78 (13.3)	−8.93 (8.8)	0.388	−0.25 (11.9)	0.913
	VRTT	−7.35 (13.1)	−10.45 (8.9)	0.116	−2.25 (5)	
ROM hip flexion	RAGT	38.15 (15.3)	43.575 (9.7)	**0.034**	3.3 (4.43)	0.092
	VRTT	41.8 (8.25)	44.05 (6.6)	0.875	0.5 (6.1)	
ROM pelvic tilt	RAGT	6.88 (3.88)	7.5 (3.38)	0.185	0.78 (1.01)	0.901
	VRTT	6.9 (3.15)	7.55 (5.85)	0.552	1.35 (4.15)	
ROM pelvic obliquity	RAGT	7.4 (3)	9 (3.025)	0.084	1.2 (2.8)	0.327
	VRTT	7.8 (3.3)	9.1 (3.9)	0.916	0.6 (4.3)	
ROM pelvic rotation	RAGT	0.44 (0.18)	0.53 (0.16)	**0.041**	0.12 (0.16)	0.277
	VRTT	0.5 (0.17)	0.5 (0.16)	0.650	0.02 (0.24)	

**Table 5 life-11-00548-t005:** Number of respondent to treatment in RAGT and VRTT groups, in terms of those outcome measures for which the minimal clinical important difference (MCID) was available in the literature. The number of patients showing improvement (I), stability (S) or worsening (W) are reported as I/S/W. ROM: Range of motion; RAGT: Robot-Assisted Gait Training; VRTT: Virtual Reality Treadmill Training; 6MWT: 6 min walking test; GMFM: Gross Motor Function Measure. ‡ Pearson Chi-squared.

	RAGT (N = 15)	VRTT (N = 15)	*p*-Value ‡
6MINWT	8/6/1	6/8/0	0.463
GMFM-88	7/6/0	3/11/0	0.081
GMFM-D	8/5/0	4/10/0	0.085
GMFM-E	9/4/0	7/7/0	0.310
Step length	1/11/0	1/12/0	0.953
Maximum ankle flexion	4/8/0	6/7/0	0.513
Minimum ankle flexion	3/9/0	4/9/0	0.748
ROM ankle flexion	2/9/1	3/10/0	0.545
Maximum knee flexion	4/6/2	1/11/1	0.168
Minimum knee flexion	1/7/4	0/11/2	0.284
ROM knee flexion	5/7/0	3/9/1	0.425
Maximum hip flexion	3/7/2	1/9/3	0.494
Minimum hip flexion	7/5/0	3/8/2	0.119
ROM hip flexion	4/7/1	3/9/1	0.838

## Data Availability

The authors confirm that the data supporting the findings of this study are available within the article. The complete raw data that support the findings of this study are available at 10.5281/zenodo.4630814.

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
