# Peer review of "The Effectiveness of Robot- vs. Virtual Reality-Based Gait Rehabilitation: A Propensity Score Matched Cohort"

_life, 2021, doi:10.3390/life11060548_

Round 1
Reviewer 1 Report
In this paper, the authors aim to compare the effectiveness of two interventions (i.e. RAGT and VRTT) for the gait rehabilitation of children and adolescents suffering from ABI. The paper is easy to read and insightful.
Introduction
The introduction provides an overview of the state of the art of the topic, what has already been done and what is the aim of the paper. To make the introduction complete, I would include a reference to the fact that these technologies are used for many categories of patients by referring to several studies reported in the literature such as:
Maranesi et al., Effectiveness of Intervention Based on End-effector Gait Trainer in Older Patients With Stroke A Systematic Review. Journal of the American Medical Directors Association, 2019, S1525-8610(19)30750-9. DOI: 10.1016/j.jamda.2019.10.010
Chen et al., A review of lower extremity assistive robotic exoskeletons in rehabilitation therapy. Crit Rev Biomed Eng 2013;41:343e363.
Methods
This session is well written and structured. The only note is to insert a precise explanation on the meaning of the GDI.
Results/Discussion
These two parts should both be improved.
The results reported in this way risk confusing the reader. In fact, the differences seen in Tables 3, 4, and 5 could be related to the fact that the pathway in the selected age group (4-12 years) is very different (as has been reported in the discussions). So I would like to ask: does it make sense to report the results by mixing all participants or would it be better to select the study participants better by focusing on a smaller age group? should the results be stratified by age?
Minor comment
Lines 319-310: fix the layout.
Insert the meaning of the acronyms under each table.
Author Response
We thank the reviewer for his/her valuable comments. We have revised the manuscript according to your comments. You can find a point-by-point rebuttal below and changes implemented within the text highlighted in yellow.
We have also corrected Table 2 (that in the previous version had a missing column due to changes to the format). It shows the baseline measures of the matched cohorts.
In this paper, the authors aim to compare the effectiveness of two interventions (i.e. RAGT and VRTT) for the gait rehabilitation of children and adolescents suffering from ABI. The paper is easy to read and insightful.
Thanks
Introduction
The introduction provides an overview of the state of the art of the topic, what has already been done and what is the aim of the paper. To make the introduction complete, I would include a reference to the fact that these technologies are used for many categories of patients by referring to several studies reported in the literature such as:
Maranesi et al., Effectiveness of Intervention Based on End-effector Gait Trainer in Older Patients With Stroke A Systematic Review. Journal of the American Medical Directors Association, 2019, S1525-8610(19)30750-9. DOI: 10.1016/j.jamda.2019.10.010
Chen et al., A review of lower extremity assistive robotic exoskeletons in rehabilitation therapy. Crit Rev Biomed Eng 2013;41:343e363.
As suggested by the Reviewer we have added the following references that underline the use of RAGT in adults with neuromotor disorders in the Introduction:
- Maranesi E, Riccardi GR, Di Donna V, Di Rosa M, Fabbietti P, Luzi R, et al. Effectiveness of intervention based on end-effector gait trainer in older patients with stroke: a systematic review. Journal of the American Medical Directors Association 2020;21(8):1036-1044.
- Mehrholz J, Thomas S, Kugler J, Pohl M, Elsner B. Electromechanical‐assisted training for walking after stroke. Cochrane database of systematic reviews 2020(10).
- Yeh S, Lin L, Tam K, Tsai C, Hong C, Kuan Y. Efficacy of robot-assisted gait training in multiple sclerosis: A systematic review and meta-analysis. Multiple sclerosis and related disorders 2020;41:102034.
- Fang C, Tsai J, Li G, Lien AS, Chang Y. Effects of robot-assisted gait training in individuals with spinal cord injury: a meta-analysis. BioMed research international 2020;2020.
Methods
This session is well written and structured. The only note is to insert a precise explanation on the meaning of the GDI.
We have included the following paragraph to explain better the meaning of the GDI:
“The GDI was developed and validated by Schwartz and Rozumalski in 2008 [32]. It is defined as the scaled distance between 15 gait feature scores (selected as those that explain the 98% of data) for a subject and the average of the same 15 gait feature scores for a control group of typically developing children. Therefore, the GDI provides an overall assessment of the deviation from a physiological gait pattern.”
Results/Discussion
These two parts should both be improved.
The results reported in this way risk confusing the reader. In fact, the differences seen in Tables 3, 4, and 5 could be related to the fact that the pathway in the selected age group (4-12 years) is very different (as has been reported in the discussions). So I would like to ask: does it make sense to report the results by mixing all participants or would it be better to select the study participants better by focusing on a smaller age group? should the results be stratified by age?
We would like to thank the Reviewer for this comment. The approach proposed is valuable, age stratification is needed to assess rehabilitation effectiveness on more homogenous group of subjects. Unfortunately, after the propensity score matching algorithm, which matched the two groups not only in terms of age but also in terms of time from injury, motor impairments, aetiology and GMFCS (see Table 2), the sample size became too small (N=15 for each intervention group) to perform a subgroup analysis. However, in order to highlight the importance of age stratification, we added a limitation of the study and we rephrased the conclusive following sentence to suggest age stratification for future works:
“Finally, the range of age of the participants was quite broad and the small sample size did not allow to perform age stratification.”
“Future works will use larger sample size, will compare the intervention effect at different ages and will investigate long-term benefits.”
Minor comment
Lines 319-310: fix the layout.
Fixed
Insert the meaning of the acronyms under each table.
Thanks, we inserted the missing acronyms in each Table caption that is above the Table, as suggested by the Journal format.
Reviewer 2 Report
The work is based on unique data obtained in a specialized well-equipped laboratory. From an ethical point of view, the design of a prospective randomized study is somewhat problematic. From this point of view, the design of the study is acceptable. On a relatively small set, a large number of parameters are monitored without determining the primary end-point. No correction was made for multiple comparisons. For this reason, the statement about the statistical significance of differences in individual parameters between groups is not justified. This is the main weakness of the presented work.
Author Response
We thank the Reviewer for his/her valuable comments. We have revised the manuscript according to your comments. You can find a point-by-point rebuttal below and changes implemented within the text in yellow.
We have also corrected Table 2 (that in the previous version had a missing column due to changes to the format). It shows the baseline measures of the matched cohorts.
The work is based on unique data obtained in a specialized well-equipped laboratory. From an ethical point of view, the design of a prospective randomized study is somewhat problematic. From this point of view, the design of the study is acceptable.
Thanks
On a relatively small set, a large number of parameters are monitored without determining the primary end-point.
We thank the Reviewer for this comment. We have now clarified that the primary outcome was the GMFM-88, which evaluates patients’ gross-motor abilities. We modified the Methods accordingly:
“Participants underwent a motor assessment before (T0) and at the end of the treatment (T1), which included the following outcome measures: Gross Motor Function Measure-88 (GMFM-88), which was selected as primary outcome, 6 minutes walking test distance (6MWT), Gillette Functional Assessment Questionnaire (FAQ) and 3-Dimensional Gait Analysis (GA).”
as well as in the Results Section (see also the new Figure 4):
“Figure 4 shows GMFM, 6MWT and FAQ, in the matched groups, before and after treatment. Both groups presented statistically significant improvement for the primary outcome, with the GMFM-88 increasing in RAGT group (Wilcoxon signed rank test p= 0.003) as well as in VRTT group (p=0.009).”
No correction was made for multiple comparisons. For this reason, the statement about the statistical significance of differences in individual parameters between groups is not justified. This is the main weakness of the presented work.
Due to the small sample size, we preferred to use non-parametric statistical tests. We compared the time factor (T0 vs T1, 2 time points), for the two interventions groups separately, using the Wilcoxon signed rank test. Since only two time points of assessment were available, we did not perform any correction for multiple comparisons. In order to evaluate the group effect, we evaluated the T1-T0 change difference for each subject and outcome measure and then we compared the group effect using the Mann-Whitney U test. Since there were only two intervention groups, no correction for multiple comparisons was performed.
However, as stated in the Discussion section, we did not find any differences in individual outcome measures between groups.
Reviewer 3 Report
The paper is well prepared and up to standard. The method of research is presented in a clear way. This paper compares the effectiveness of two interventions (i.e. RAGT and VRTT) for the gait rehabilitation of children and adolescents suffering from ABI. The results are interesting and useful. However, I would like to suggest the authors to further extend the details in the section 5 conclusion on page 12 to describe more about the contribution of this research work.
Author Response
We thank the reviewer for his/her valuable comments. We have revised the manuscript according to your comments. You can find a point-by-point rebuttal below and changes implemented within the text highlighted in yellow.
We have also corrected Table 2 (that in the previous version had a missing column due to changes to the format). It shows the baseline measures of the matched cohorts.
The paper is well prepared and up to standard. The method of research is presented in a clear way. This paper compares the effectiveness of two interventions (i.e. RAGT and VRTT) for the gait rehabilitation of children and adolescents suffering from ABI. The results are interesting and useful. However, I would like to suggest the authors to further extend the details in the section 5 conclusion on page 12 to describe more about the contribution of this research work.
We thank the reviewer for his/her valuable comments. We have expanded the Conclusion section describing more about the contribution of our work. Specifically the following paragraph was added:
“Recently, more and more rehabilitation technologies are on the market. Each of them has its indication for use but this is often too generic, e.g. for gait rehabilitation and does not provide specific indications on target users. Thus, it is difficult for a clinician to choose the best device for each patient. This work would like to contribute in this direction. The approach used and the results obtained, although preliminary, pave the way for the definition of guidelines for the treatment of children and adolescents suffering from ABI.”
Reviewer 4 Report
In general, a description and writing style of the paper are of a good quality. However, the presentation of research methodology and experimental data analysis must be improved. E.g. the algorithms must be presented in a flow-chart form, all acquired experimental data (essential ones) need to be presented via plots and charts. As is, it is not clearly articulated the influence of the studied technology interventions (RAGT and VRTT) in gait rehabilitation of patients, e.g. different age groups. Explicitly presented comparative analyses on effects of these technologies are missing.
Author Response
We thank the reviewer for his/her valuable comments. We have revised the manuscript according to your comments. You can find a point-by-point rebuttal below and changes implemented within the text in yellow.
We have also corrected Table 2 (that in the previous version had a missing column due to changes to the format). It shows the baseline measures of the matched cohorts.
In general, a description and writing style of the paper are of a good quality.
Thanks
However, the presentation of research methodology and experimental data analysis must be improved. E.g. the algorithms must be presented in a flow-chart form.
Algorithms used are made of simple commands and flow-charts would have been bare. Therefore, we preferred to improve the text of the manuscript. We added the following sentence to section 2.4.1:
“The PSM algorithm was developed in Rstudio by means of the MatchIt library. The matchit() function was used with the method “nearest” to implement the 1:1 nearest-neighbor matching.”
We also better described the Matlab software used to extract gait parameters (section 2.4.2):
“Kinematic curves were analyzed in Matlab by using an ad hoc algorithm designed to extract, for the right and left leg, the foot progression in stance, maximum and minimum flexion angle and the range of motion (ROM) in the sagittal plane for ankle, knee and hip, and the ROM of pelvic tilt, obliquity and rotation.”
However, the presentation of research methodology and experimental data analysis must be improved. E.g. [] all acquired experimental data (essential ones) need to be presented via plots and charts.
Authors are aware that figures often provide more readable and immediate results. Therefore, Table 3, which reported the results in terms of functional measures, was transformed in Figure 4 and the text was changed accordingly as follows:
“Figure 4 shows 6MWT, FAQ, and GMFM in the matched groups, before and after treatment. Both groups presented statistically significant improvement for the FAQ (Wilcoxon signed rank test p= 0.017 for RAGT and p=0.046 for VRTT), the GMFM-88 (Wilcox-on signed rank test p= 0.003 for RAGT and p=0.009 for VRTT), and its dimensions D (Wilcoxon signed rank test p= 0.005 for RAGT and p=0.018 for VRTT) and E (Wilcoxon signed rank test p= 0.002 for both RAGT and VRTT). The percentage of patients with clinically relevant changes in GMFM-88, GMFM-D and GMFM-E were 54%, 62% and 69% in the RAGT and 21%, 29% and 50% in the VRTT group. Nobody experienced a worsening in his/her gross motor abilities. The 6MWT improved significantly in the RAGT group (Wilcoxon signed rank test p= 0.003; 53% of patients with clinically relevant changes and 7% of patients with a worsening above MCID) and had a trend of improvement (p=0.056) in the VRTT group (43% of patients with clinically relevant improvements and 0% with clinically relevant worsening). No differences in the therapy effect were found as demonstrated by the Mann-Whitney U test (all p-values>0.070).”
However, due to the huge amount of data, we decided to keep the presentation of the results of the other outcome measures in tables. Specifically, Tables report median and interquartile values for each outcome measure and intervention group as well as changes T1-T0 of each group. P-values of the statistical analysis are also reported.
As is, it is not clearly articulated the influence of the studied technology interventions (RAGT and VRTT) in gait rehabilitation of patients, e.g. different age groups. Explicitly presented comparative analyses on effects of these technologies are missing.
The comparative analysis between technologies is reported as pre-post changes computed for each group/treatment. The results were clarified, and the following sentence in the Discussion section was rephrased to discuss more about the differences between the two treatments.
“Data analysis performed on the matched cohorts revealed that gross motor abilities significantly improved in both groups; however, specifically considering the GMFM-88 and GMFM-D there is a trend of higher percentage of patients in the RAGT group that gained a clinically relevant change. Results also showed that similar percentages of patients in each group improved their endurance and their step length above MCID, even though the improvements were statistically significant only for the RAGT group. Therefore, patients who underwent RAGT treatment had a slightly higher functional gain. For what concern gait analysis, data confirmed a beneficial intervention of RAGT at proximal level (i.e. pelvis and hip) and a positive effect of VR on distal districts (i.e. foot and ankle) and on the overall gait pattern quality. Both treatments barely worked on knee joint. The statistical analysis did not find any differences between the two interventions.”
Finally, for what concern different age group, we are aware that age stratification is needed to assess rehabilitation effectiveness on more homogenous group of subjects. Unfortunately, after the propensity score matching algorithm, which matched the two groups not only in terms of age but also in terms of time from injury, motor impairments, aetiology and GMFCS (see Table 2), the sample size became too small (N=15 for each intervention group) to perform a subgroup analysis.
However, in order to highlight the importance of age stratification, we added a limitation of the study and we rephrased the conclusive following sentence to suggest age stratification for future works:
“Finally, the range of age of the participants was quite broad and the small sample size did not allow to perform age stratification.”
“Future works will use larger sample size, will compare the intervention effect at different ages and will investigate long-term benefits.”
Round 2
Reviewer 1 Report
Thank you for your detailed responses. I have no further comments.
Reviewer 4 Report
One more careful revision would be helpful to remove or edit any obscure expressions throughout the whole manuscript.